# Shaped-Charge Learning Architecture for the Human–Machine Teams

**DOI:** 10.3390/e25060924

**Published:** 2023-06-12

**Authors:** Boris Galitsky, Dmitry Ilvovsky, Saveli Goldberg

**Affiliations:** 1Knowledge-Trail, San Jose, CA 93635, USA; 2Computer Science Faculty, HSE University, Moscow 101000, Russia; 3Department of Radiology at Massachusetts General Hospital, Boston, MA 02114, USA; savelig@gmail.com

**Keywords:** machine-learning support for human–machine teams, deep and nearest-neighbor learning, structural entropy production, maximum entropy production

## Abstract

In spite of great progress in recent years, deep learning (DNN) and transformers have strong limitations for supporting human–machine teams due to a lack of explainability, information on what exactly was generalized, and machinery to be integrated with various reasoning techniques, and weak defense against possible adversarial attacks of opponent team members. Due to these shortcomings, stand-alone DNNs have limited support for human–machine teams. We propose a *Meta-learning/DNN → kNN* architecture that overcomes these limitations by integrating deep learning with explainable nearest neighbor learning (kNN) to form the object level, having a deductive reasoning-based meta-level control learning process, and performing validation and correction of predictions in a way that is more interpretable by peer team members. We address our proposal from structural and maximum entropy production perspectives.

## 1. Introduction

Despite much advancement in AI systems to reproduce human language, designing agents that use natural language (NL) to communicate with human agents in real-world productive environments remains a major challenge. A major long-term goal for Artificial Intelligence (AI) is to build machines capable of conversing, interacting, planning, agreeing, and disagreeing with humans in NL. Although much progress has been made in language models that imitate human language, effective collaborating agents must go beyond this by understanding and communicating their beliefs, goals, and intentions of themselves and their peers; and by planning joint actions that account for their peer’s goals and intentions. In terms of supporting such collaboration, the machine-learning (ML) engines of these agents need to combine the efficiency of deep-learning networks (DNN) and the transparency of classical AI methods such as deductive reasoning and deterministic inductive learning.

This is the first paper in a series of two with a focus on the architecture and methodology, while the second paper addresses applications and evaluation. The purpose of such a split is to convince the reader that the novel learning architecture is fruitful and efficient, as seen analytically, without the involvement of an evaluation as a black box. This is to oppose our approach to the most popular papers from the DNN community, where the evaluation section is frequently essential, and an approach cannot be accepted by the community without it.

As team performance can be characterized by entropy [1], we proceed from unexplainable ML support vulnerable to adversarial attacks to achieve explainable, collaboration-efficient support [2,3,4,5]. Entropy evolves accordingly from high structural entropy to a reduced one under higher team efficiency and productivity. 

We build a hybrid ML structure, starting from neural gradual descent, which culminates in nearest-neighbor learning. We attempt to combine the best of both worlds: accuracy of neural learning with explainability and meaningfulness of nearest neighbor learning and reasoning, which validates and corrects DNN results. For a given prediction, we first apply neural gradual descent, and then as we derive a candidate prediction, we apply nearest neighbors to it to produce an exact, interpretable result. Hence, the prediction sessions of shaped-charge ML ‘explode’ when the final stage is achieved and after a set or sequence of relevant samples is applied.

We address not only such limitations of stand-alone DNN as a lack of explainability and demonstration of explicit generalization, leveraging common sense, vulnerability to adversarial attack, and a lack of introspection. Some of these limitations can be overcome by wrapping a DNN with an explainable ML component [6], such as nearest neighbor learning (kNN), and the other by meta-learning in the form of reasoning about the DNN + kNN learning process. 

One implementation of shaped-charge ML is question answering based on a neural machine-learning subsystem followed by a syntactic match with the candidate answer (a match is characterized by minimum entropy), which verifies and corrects it when needed. A similar approach is used for summarization: the kNN verifies if, for each summary phrase, there is a similar sentence in the text being summarized. Another architecture included is a transformer-based content generation that is verified and corrected via a web mining fact-checking component. 

Combining symbolic reasoning and numeric processing is known to be a promising direction in AI and applications. Our hybrid architecture can be expressed as *Meta-learning/DNN → kNN:kNN* follows *DNN*, and learning configuration is controlled in the metalanguage of inductive learning, such as inductive logic programing (ILP) or discourse analysis as a meta-language for object-level semantics. One way to look at this research is building a discourse theory around DNN learning as if all learning occurs in a text. For other kinds of data, such as images, we consider discourse in a broader context at an abstract meta-level.

In this work, instead of a classifier ensemble approach that would combine high-performing DNN and lower-performing but explainable kNN, we apply kNN to an enriched DNN prediction instead of the original data being classified. We frame the results by structural or process costs, where the least entropy expended is optimal [1], and by effectiveness, where maximum entropy productivity is the most effective outcome, i.e., an “ML explosion”.

DNNs are architectured to acquire a hierarchical set of representations of text or images. These representations embed the input data in increasingly abstract spaces, which gradually become abstract enough for the problem to be solved, such as classification or generation, prior to being solved [7]. We ascend all the way to a meta-learning level controlling the structure of DNN processors operating at the object level.

Hence, we extend DNN with kNN for explainability and accuracy verification, cover it with meta-learning to control both learning and prediction processes, and enable the overall system with reasoning capabilities. For applications in a text domain, we implement meta-learning in the form of textual discourse. Syntax and semantics, together with DNN recognition of syntax and semantics, constitute the object level, while the discourse forms the meta-level, controlling this object level.

Our first section provides examples of failures of some human–machine teams which motivate this study. Applications of shaped-charge learning are presented in the companion paper of this volume (“Applications of shaped-charge learning”).

### 1.1. Examples of Failures of Human–Machine Teams

Team failures represent wasted maximum entropy structure costs and minimum entropy productivity or effectiveness.

In November 2022, Meta (Facebook) gave access to its new language model (LM) called Galactica, intended to help researchers. However, instead of earning the great success that Meta was expecting, Galactica was shut down after three days of intense criticism. A main flaw with Galactica is that it is not able to differentiate true from false facts. Users were receiving fake papers attributed to real authors. For example, Galactica produced Wiki articles about the history of animals in space. Users can identify misrepresentations when they involve entities such as space bears while having difficulties with a subject they may not be familiar with.

The exaggeration of DNN usability is a big problem in the self-driving auto domain. Tesla has been criticized by the Department of Motor Vehicles in California, USA, when it made claims about its autopilot and self-driving features (which are based on DNNs). These claims are believed to be deceptive. Tesla is at risk of having its licenses to operate as a vehicle manufacturer and auto dealer in California revoked. Tesla has also been criticized for how it positions its Advanced Driver Assistance System. One of the main concerns has been the actual names of the systems: Autopilot and Full Self-Driving Capability. Some people believe the names suggest that the systems are autonomous, even though they are only driver-assisted systems.

The field of DNN is especially subject to fake claims as the technology looks magical, but it is hard for a general audience to understand how it works. Many businesses that claim to have developed DNN algorithms for images or texts do not always disclose that human agents are acting behind the scenes. This tradition of hiding human input in DN systems is a known “secret” in the AI business community. MMC Ventures found in 2019 that almost half of all startups claiming to use AI technology are not actually using AI in their products.

Another example of a failure of human–machine teams is in the CRM (customer relationship management) domain, where a subpar performance of a business, such as a call center, is due to faulty ML support. The concept of Distributed Incompetence (DI), opposite to Distributed Knowledge, was discovered [8]. In a business overwhelmed with DI, a team of customer support employees is managed in a way that, being rational, impresses a customer but with total irrationality and incompetence and a lack of capability to get things done. In most cases, the whole business or a particular team member gains from DI by means of refusing monetary compensation to customers who received faulty products or services. DI has been identified in a variety of organizations, and its commonality, together with specific DI features, has been analyzed. A DI rate has been assessed in financial organizations, and a solution to tackle it based on conversational AI has been proposed.

Distributed Incompetence is frequently connected with misrepresentations. When a team is eager to achieve a goal, it can ignore rationality by dissembling and lying. In most cases, such a team lacks competence in what they attempt to do. Nonetheless, team members intensely try to convince their peers of the opposite. This is aggressive Distributed Incompetence.

There have always been impressive stories of lies and deceit, tales about founders sharing partial truths about how their companies were founded and which products were developed, and CEOs exaggerating the features of their products to mislead an audience. Some CEOs misrepresent the number of users (such as Twitter); some provide misrepresentations to Congress concerning privacy issues, assuring they have full control over the personal data of their users (Facebook). One of the most striking stories is that of Elizabeth Holmes, the founder of Theranos, who developed blood test technology [9]. 

Over the last half-decade, the first and second authors leveraged Theranos’ texts as a dataset for argumentation analysis. Texts from the Wall Street Journal with claims that the company’s conduct was fraudulent were subjected to discourse analysis. The authors were developing argumentation mining and reasoning algorithms [10] witnessing the Theranos story, and obtaining content from the Theranos website. A part of the audience believed that the case was initiated by Theranos’ competitors, who felt threatened by the proposed blood test technique of Theranos. However, our argumentation mining demonstrated that Theranos’ claims were faulty. The SEC (2018) [11] states that Holmes raised a few hundred million from investors while making false statements about Theranos technology.

We conclude this section with the statement that teams need better ML systems than those that are currently available, especially with reasoning capabilities. In terms of entropy, a maximal structural entropy indicates that energy has been wasted, meaning a minimum entropy in effectiveness has been achieved. Our major direction of improvement is a hybrid ML plus reasoning architecture. As a hybrid DNN plus reasoning architecture being suggested, we analyze the shortcomings of a DNN first and then that of stand-alone logic-based learning such as Inductive Logic Programming second. 

### 1.2. Limitations of Stand-Alone DNN

AI in general and DNN in particular are important in our lives, having made a huge impact in a broad range of domains, including medical imaging, process controls, navigation, and language modeling. Carefully designed feature engineering used in traditional ML, such as classification and pattern recognition systems, is not always scalable for complex problem domains and big data. In many domains, depending on how sophisticated the task is, DNN can outperform shallow neural networks that prevented fast training and building hierarchical models of complex training data [12]. However, DNNs have strong limitations in human–machine environments in comparison with traditional logical AI.

In general, any task that requires reasoning (including software development programming, leveraging the scientific method, or planning and algorithmic data processing) is beyond the DNN skillset. Even learning a sorting algorithm is extremely difficult for DNN since it is a chain of continuous topological transformations of one representation space into another. DNNs are capable of mapping one data representation, D1, into another data representation, D2, if a learnable continuous transformation from D1 to D2 exists and if having the training data in D1–D2 format is available. Although a DNN model can be viewed as a sequence of instructions, most programs cannot be encoded as DNNs. For most problems, either a respective reasonably sized DNN that solves the task does not exist, or even if it does exist, it may not be learnable, i.e., encoded topological transformations may be overly complex, or there may be a lack of data to train it [13].

Boosting current DNN architectures by forming additional layers and involving a higher amount of training data can only superficially alleviate some of these limitations. Scaling up DNNs is not expected to solve the essential limitation of the DNNs that are limited in what they can represent and that most of the algorithms that need to be learned cannot be expressed as a continuous topological transformation. Although DNNs are good at image recognition, machine translation, and malware detection, their use is often critiqued for their lack of robustness in adversarial settings and a lack of capability to back up their predictions.

One of the risks of neural AI is that of misinterpreting what DNN models do and overestimating their abilities. A fundamental feature of the human mind is the “theory of mind”, the human ability to reason about intentions, beliefs, and knowledge of subjects in the real world and themselves (introspection). Applied to a DNN, this is interpreted that when a model is trained to generate captions to describe pictures, we expect this model to “understand” the contents of the pictures, as well as the captions it generates. “Understanding” here is, in a sense, mapping the real world into a symbolic representation of it. DNN users are then astonished when any deviation from the type of images in the training dataset leads a DNN to yield meaningless text for images.

We now enumerate the limitations of DNN confirmed by an expert community [14] and how the shaped-charge approach can overcome them:

*Representation of the environment.* An insufficient capability exists to form representations of the real world with a high level of generality to allow easy and efficient manipulation of representations. We will apply Inductive Logic Programming when/if users request an explicit generalization result. For question answering, we will show syntactic and semantic generalizations as a measure of answer relevance. 

Abstract concept formulation. A limited ability to comprehend, manipulate, and formulate abstract concepts to reduce the high dimension of a stimulus or express the meaning of an NL expression. The integration of a DNN with a formal concept analysis [15] in chatbot dialogue management helped to alleviate this shortcoming. 

*Causal links.* A limited ability to identify the causal direction in which features lead to which other features and a lack of skill in the generalization of learned causal relations. Combining induction and deduction is essential to reason causal relationships; these relationships are also essential to team collaboration. 

*Meta-reasoning and meta-learning.* Limited acquisition skills in how to learn and how to introspect. These are reflected by difficulties in selecting which algorithmic adaptation learns optimally in each domain-specific learning task, leveraging the meta-data about the learning tasks.

DNN systems do not really “understand” their input, in any sense, from the human standpoint. Human representation of images, sounds, and language, as well as tactile feeling, is grounded in the sensory experience of humans [13]. ML is unable to rely on “human sensor” experiences and thus cannot “understand” its inputs. By digesting large numbers of training examples, DNNs learn a topological transformation that maps data to human concepts for this specific set of examples. However, this topological mapping is just an approximation of the original model in human minds, yielding from human experience as embodied agents (Figure 1). Hence, it is necessary to wrap a DNN by some formalization of human sense.

DNNs are often criticized for their poor performance in adversarial settings and a lack of skills to make predictions rationally. Papernot and McDaniel [16] leverage the DNN structure to improve learning-based reasoning and achieve robust and interpretable decision methodologies. The authors introduce the Deep k-Nearest Neighbors that combine the kNN with data distribution learned by each DNN layer. A test input is matched with its neighboring training samples with respect to the distance between these samples in the DNN layer representations. Ref. [16] demonstrate that the confidence estimates of the labels of these points obtained by kNN can be made outside the training set’s vicinity. A kNN is used to estimate a lack of support for a prediction in the training data. Confidence levels can also be computed for malicious inputs such as adversarial examples; this assures stability in performance with respect to inputs that are outside the DNN “understanding”. A kNN component also supports human-interpretable explanations of predictions. Papernot and McDaniel apply the Deep k-Nearest Neighbors algorithm to several datasets to demonstrate that the confidence estimates properly compute inputs outside the training dataset. Explanations produced by a kNN are comprehensive and assist in handling DNN failures.

Analogical reasoning is another domain essential for team support. DNN can imitate it only when somewhat similar reasoning chains are available for training. As a result, when asked the same ‘free fall’ or ‘ice in the water’ question, GPT-3 returns the wrong answer: “The heavier object will fall faster”. Due to their lack of grounded reasoning, current LMs also have issues with truthfulness [17] and factuality [18], which we addressed in our earlier studies [19].

Having analyzed the limitations of a DNN, we now proceed to the analysis of shortcomings of an approach that is expected to be complementary: inductive logic programming (ILP), which is designed for learning and reasoning (natural integration with other reasoning components) at the same time.

### 1.3. Limitations of Stand-Alone Inductive Learning

In comparison to most ML approaches, Inductive Logic Programming has several advantages, primarily by being able to generalize from a small number of samples, sometimes even from a single example. Given a set of clauses for a target predicate and background knowledge, the ILP problem is formulated as an induction of a hypothesis that correctly generalizes the available examples with the background knowledge. A key characteristic of ILP is that it contains the samples, background knowledge, and hypotheses as logic programs. These programs include sets of logical rules in the form of clauses. The feature of an ML system to have team members review generalizations is essential for successful applications. Because hypotheses are logic programs, they can be read by humans, which is essential for explainable AI. Being symbolic systems, ILP naturally supports lifelong transfer learning [20] and learning for team support.

The fundamental limitation of ILP is a slow search in the extended space of a hypothesis. One possibility is to use a set covering algorithm to acquire the hypothesis one clause at a time. Such algorithms are often efficient because they are based on examples [21]. However, these ILP approaches sometimes over-generalize; they learn overly specific solutions and experience difficulties learning programs with nontrivial flows that are recursive [22]. An ILP would benefit from control at the meta-level of how deeply generalized clauses are and which clause structures to maintain. In this study, we will apply such a meta-level control to a DNN and a kNN/ILP working together.

Another approach is to encode the ILP as answer set programming. Such algorithms frequently acquire optimal and recursive programs efficiently and serve as a basis for state-of-the-art answer set programming solvers but are limited in performance in large domains with extensive background knowledge [23].

### 1.4. A Promise of Hybrid Architecture

We now proceed to the approaches which inspired our hybrid *Meta-learning/DNN → kNN* architecture. We propose to rely on gradient descent to link a DNN and kNN (Section 3), and we will have a DNN and kNN controlled by meta-learning further developed in Section 4. A kNN can be potentially substituted by an ILP component: its hypotheses can be formed as DNN predictions.

Due to the success of DNNs, one of the main integrated techniques is to combine them with logical reasoning, which is called neuro-symbolic computation [24,25]. The main goal is to establish a unified framework that can make flexible approximations using DNNs and perform tractable and multi-hop reasoning using first-order logic. In this study, such a framework is based on a sequential *DNN → kNN* hybrid controlled in meta-language.

Inductive Logic Programming (ILP) [26] is a sound formalization for finding theories from given examples using first-order logic as its language [27]. Ref. [28] introduce an approach from the ILP family called “learning from failures” where a problem is split into three separate stages: generate, verify, and constrain. In the generate stage, the learner generates a hypothesis (a logic program) that satisfies a set of syntactic constraints. In the verify stage, ILP verifies how training examples satisfy the hypothesis. A failure of the hypothesis means that it does not yield all of the desired positive examples or it yields a negative example. In the case of a hypothesis failure, at the constraint stage, ILP acquires hypothesis constraints from the failed hypothesis to reduce the hypothesis space. Such a reduction constrains the hypothesis to be generated in further iterations.

ILP algorithms use syntactic bias, which forces syntax constraints on hypotheses, such as the number of variables allowed in a clause, and also semantic bias, which reduces the number of hypotheses based on their semantics, such as whether they are irreflexive or functional.

Metarules *control* syntactic bias used by many ILP approaches [29], including Metagol and ∂ILP [30]. A metarule is a higher-order clause that defines the exact form of clauses in the hypothesis space in an ILP and defines the learning configuration for an arbitrary learner, such as a DNN. For instance, the chain metarule is of the form *l(A,B) ← m(A,C),n(C,B),* where *l, m,* and *n* are predicates, *A, B,* and *C* denote predicate variables, and the result allows for instantiated clauses such as *final_processing_step(A,B):- reorder(A,C), first_processing_step(C,B).* A human team member must either supply a set of metarules or rely on a set of metarules constrained to a specific fragment of logic, such as dyadic Datalog [31] for ILP. 

One implementation of shaped-charge ML is question answering based on a neural machine-learning subsystem, first followed by a syntactic match with a candidate answer. Another architecture is a transformer-based content generation that is corrected by a web mining fact-checking component. 

A boost of performance in large language models can be facilitated by one of the following approaches: (1)Increasing the size of the models in both depth and width;(2)Enlarging the number of tokens that the model was trained on;(3)Building cleaner datasets from more diverse sources;(4)Improving model capacity through sparsely activated modules.

These improvements include maintaining the size of a model and token set, corpus consistency, and completeness, as well as sparse model activation, which can be assessed and controlled from the meta-level. Instead of intervening by a team member, these responsibilities are automated in the form of meta-rules.

Combining DNN and explainable ML approaches, which both implement gradient descent [30], addresses a DNN’s lack of explainability of predictions and assures a “smooth” transition from the former to the latter in the course of a prediction session. The knowledge that deep networks are effectively path kernel machines can significantly improve interpretability. The weights of synaptic connections in a DNN have a direct interpretation as a superposition of the training samples in a gradient space, where each sample can be mapped into the corresponding gradient of the model. Combining deep and nearest-neighbor learning under the single gradient descent umbrella leverages both the high-level performance of the former and the interpretability of the latter. Moreover, a meta-learning control automates routines associated with feature and training datasets engineering. 

Evans and Grefenstette [30] proposed Differentiable Inductive Logic Programming (∂ILP), which is an environment for building logic programs from given samples relying on differentiation. The ∂ILP framework reduces an ILP to an optimization process that can be solved via gradient descent. Its differentiability establishes a promising merge of ILP and neural networks to deal with sub-symbolic and noisy data.

There are the following bottlenecks for learning complex programs and structured data: (1)The number of clauses grows;(2)A high number of ground atoms can be generated with function symbols;(3)As the search space grows, the computational costs and space required increase quadratically.

Shindo et al. (2021) [32] resolve these issues by proposing a new differentiable algorithm for acquiring logic programs that merge adaptive symbolic search and continuous optimization methods.

Hence, shaped-charge learning functions as follows: a kNN completes an operational DNN, and meta-reasoning controls them both by verifying and correcting. DNN models and kNN cases are domain-specific, but meta-reasoning is not. Meta-reasoning controls active learning, the current dataset, generalizes cases of failure of object level, decides which ontologies to involve, and controls how the whole ML agent communicates with its peers. Explanation chain rules are also parts of the meta-level, along with rules extracted by an ILP.

The contribution of shaped-charge learning in comparison with other hybrid *DNN + logical* architectures is as follows:(1)We rely on kNN for explainability as it is the most simple and intuitive representation of learning via similarity, applicable for both metric learning spaces and structures. For structured learning, similarity is defined as a cardinality measure of maximum common substructures such as subtrees (parse trees in linguistic representation).(2)While a parallel kNN + DNN configuration has been explored [16], a consecutive *DNN → kNN* architecture has not, providing a uniform framework for gradient descent in large but unexplainable followed by reduced and explainable subspaces. The gradient descent in the same space for multiple learning methods also assures higher interpretability.(3)While hybrid architectures combining unexplainable but efficient traits with explainable but reduced efficiency have been proposed, shaped-charge defines a clear boundary between which method is applied to which part of the dataset at a certain level of granularity.(4)While meta-reasoning and meta-learning have been broadly used in ML, the meta-level support of the gradient descent in *DNN → kNN* allows for better manual feature engineering, handling unbalanced datasets, out-of-distribution data, and guided active learning capabilities. Active learning supported by meta-learning then facilitates fine-tuning.

### 1.5. Background Information

K-Nearest Neighbors [33] is a supervised machine learning algorithm used for classification and regression problems. The aim is to find the K-Nearest data points in the training dataset and classify new data points according to the labels of their nearest neighbors. KNN is a non-parametric approach, which means that it does not make any assumptions about the underlying data distribution. KNN is a popular choice for many classification tasks due to its simplicity and ease of implementation. It is based on the assumption that similar data points are more likely to belong to the same class. This assumption is based on the idea of similarity measurement, which is performed using a distance metric such as Euclidean distance. KNN is also a lazy learner, which means that it does not learn a discriminative function from the training data but rather simply stores the training data and uses it as a reference when a new data point is encountered. KNN has several advantages, such as its simplicity and ability to handle large datasets. It is also robust to outliers since they have less of an impact on the average of the k-Nearest Neighbors. In addition, it is non-parametric, which means that it does not make any additional assumptions about the dataset.

Deep Neural Networks (DNNs) are a class of Artificial Neural Networks (ANNs) that are composed of multiple layers of neurons and are used for a variety of applications such as image recognition, speech recognition, natural language processing, and reinforcement learning. DNNs have become increasingly popular due to their ability to model complex nonlinear relationships between inputs and outputs. Unlike traditional ANNs, DNNs are able to learn complex functions through the use of multiple layers of neurons, allowing them to effectively represent complex data. Additionally, DNNs are capable of leveraging large datasets to learn patterns and make predictions.

DNNs have become increasingly popular in the field of natural language processing (NLP) due to their ability to learn complex relationships between words and phrases. DNNs are able to learn word embeddings, which are representations of words that capture their semantic and syntactic information. DNNs have been used in a variety of NLP tasks, such as sentiment analysis, text classification, question answering, and machine translation. DNNs have been shown to outperform traditional methods in these tasks due to their ability to capture complex relationships in data. DNNs have also been used for language modeling, which involves predicting the next word in a sentence. This has been applied to speech recognition systems and has improved their accuracy significantly. In addition, DNNs have been used for text generation, which involves generating new text from a given input. This has been used for applications such as automated summarization, automatic poetry generation, and conversation.

## 2. Extending Traditional DNN Architecture towards a kNN

DNNs learn a hierarchical set of representations, ascending to higher levels of generalizations. DNNs embed an input pattern into increasingly abstract spaces, at some point, into a space where a prediction can be made using a baseline logistic regression. It is usually implemented by a SoftMax layer which is the last one in most DNN architectures.

This layered representation allows DNNs to produce adequate generalizations on data presented to the model at test time. However, phenomena such as adversarial examples and, in particular, the ones produced by feature adversaries [34] show that language representations learned by DNNs are not as reliable as the DNN community initially estimated. This is also true for cases with a lack of invariance to translations [35]. Because DNN training algorithms rely on the assumption that test data is obtained from a similar distribution to the training data, they perform reasonably well in domains with similar distributions. However, when a DNN is used in health or security domains where safety and reliability are essential, it is necessary for a meta-learning agent to control overfitting. It becomes important to apply machinery to identify when DNN relies too much on the representations it has built from its training data, which cannot be transferred to the real world. In particular, a kNN is expected to analyze these internal representations at test time to detect contradictions with patterns extracted from the training data.

Papernot and McDaniel [16] exploit the DNN structure to enable new learning-based inference and decision strategies that achieve robustness and interpretability, introduced by Deep k-Nearest Neighbors (DkNN, Figure 2). In contrast to this study, we apply a kNN **after** a DNN to zoom in on the space of possible predictions to improve accuracy and the capability to integrate with various forms of reasoning. 

A popular method to interpret a DNN output involves looking for training samples that are close to the query in Euclidean space [36]. Path kernels propose the exact space to implement these comparisons and to relate them to the model’s predictions. 

We combine a deep neural network (on the left in Figure 2), representation outputs by each layer (in the middle), and the nearest neighbors found at each layer in the training data (on the right). Topics of *wildlife watching* and *transportation* indicate training points. Prediction accuracy is high when there is homogeneity among the nearest neighbor labels. Explainability of the outcome of each layer is also provided. A boost in accuracy is assured by identifying nonconformal predictions from the kNN results obtained for out-of-distribution, adversarial text examples encoded by distinct network layers. 

## 3. Consecutive DNN and kNN Architecture

DNNs trained by a gradient descent algorithm are similar to kernel machines in a mathematical sense. Kernel machines store the data points and leverage them directly for prediction via a distance-measure function. This nicely improves the explainability of DNN weights, as they are essentially a superposition of the training set samples. What a DNN structure does is embed knowledge of the target function into the kernel. Most ML systems and DNNs, in particular, learn using certain versions of gradient descent (GD) [37]. Starting with an initial parameter vector $w_{0}$ and a loss function $L=\sum_{i}L(y_{i}^{*},y_{i})$, GD iteratively updates the DNN weights *w* by subtracting the loss’s gradient from them, normalized by the learning rate *ϵ* as follows: $$w_{s+1}=w_{s}-\epsilon \nabla_{w}L(w_{s})$$

The iterations stop when the gradient is zero and the loss value is optimized. Learning via GD assures that its end result is almost always a kernel machine, and it is invariant with respect to the number of network layers or neuron connection structures. 

Kernel machines that implement GD rely on what we call a *path kernel*. If a learning rate is minimized, the path kernel between a pair of data samples is the integral of the dot product of the GD at the pair of respective points over the path traveled by the learning parameters as follows:$$K\left(x,x^{’}\right)=\int_{c(t)}^{}\nabla_{w}y(x)\cdot \nabla_{w}y\left(x^{’}\right)dt$$
where *c*(*t*) is the path. Informally, the path kernel indicates the distance between the pair of data points as it varies from iteration to iteration. The lower the distance between the variation for *x* and *x*_0_ is, the higher the weight of *x*_0_ in predicting *y*.

The path kernel is intended to measure the similarity between examples (Figure 3). In the 2D training case, as the weights travel through a path, the model’s GD vectors on the weight plane for *x*, *x*_1_, and *x*_2_ are updated. The kernel *K*(*x*, *x*_1_) and *K*(*x*, *x*_2_) are then the integral of the dot product of the gradients ∇*wy*(*x*) and ∇*wy*(*x*_1_) traveling through the path. Since ∇*wy*(*x*) · ∇*wy*(*x_1_*) is greater than ∇*wy*(*x*) · ∇*wy*(*x*_2_) on average traveling through the weighted path, *y*_1_ is more important than *y*_2_ in predicting *y*, all things being equal. 

### 3.1. A Gradient Descent in Linguistic Space

We visualize the gradient descent in the linguistic space as getting closer and closer to the linguistic expression, which makes it complete and truthful (see Figure 4). The question “gradually descends” to the answer so that the similarity between them increases. We start with the sentence in a certain vicinity of the question, but not close enough; the verb is somewhat similar, but the entity is different. We then navigate through other incorrect entities until we converge on entity = *Lermontov* and predicate = *killed.* The role of verb (*x*_1_) and noun (*x*_2_) in matching a question by its answer is different. As we navigate the space of embedding a DNN, the facts are not necessarily true, but they support a monotonic convergence. Transitioning to a kNN space is an abrupt change as we now navigate through factually correct answers but not as smoothly organized as under a DNN.

### 3.2. A Gradient Descent for kNN

A regression is usually defined as an unknown function *f*: *R^D^ → R* that is predicted from the training data {(*x*_1_, *y*_1_), (*x*_2_, *y*_2_), …, (*x_N_*, *y_N_*)}, where *x_j_* is a data point, and *y* is the corresponding target value. The predicted function $\hat{f}$ is chosen to be the one that minimizes some loss function. For a classification task having T classes, this loss function is as follows:$$MSE\left(\hat{y}\right)=\sum_{t=1}^{T}\sum_{i=1}^{N}(y_{ti}-{\hat{y}}_{ti}{)}^{2}$$
where $\hat{y}$ denotes the predicted probability of point *x_i_*, and *y_i_* denotes the true label (either 0 or 1) of a point *x_i_*. For any query point *x_i_*, kNN methods produce a prediction of the value $\hat{y}$ by relying on the labels of its k nearest neighbors. In order to have a smooth boundary, each neighbor votes for the query label based on its distance from the query point.

The Nadaraya–Watson kernel for regression is expressed as follows:$$\hat{y}\left(x\right)=\frac{\sum_{j}y_{j}V_{j}}{\sum_{j}V_{j}}$$

The vote *V_j_* cast by each label around the query point ***x*** is usually chosen to be a function that decays exponentially as the distance from the query point increases, for example, a Gaussian kernel, which is expressed as follows:$$V_{j}=exp\left(\frac{-d(x,x_{j})}{2\sigma^{2}}\right)$$

Determining votes *V_j_* would rely on a well-defined distance measure, which is mostly unavailable, especially in the NLP domain. 

A kNN requires feature scaling; however, scaling all of the data to the same range is not sufficient, and when the number of features is large, it is impossible to scale the data manually. In a traditional kNN, the distance between the query object and the *j*-th object in dataset *d*(*x_q_*, *x_j_*) could be described as $d\left(X_{q},X_{j}\right)={\left\|X_{q}-X_{j}\right\|}_{2}^{2}$.

For its feature matrix *X*, the feature can be scaled using a vector *A*, thus turning the distance function *d_2_*(*x_q_*, *x_j_*) as follows:$$d\left(X_{q},X_{j}\right)={\left\|{AX}_{q}-{AX}_{j}\right\|}_{2}^{2}$$
where *d*(***x, xj***) should be replaced by a more general metric: *d_L_*(***x***,***xj***). If *L* = *A^T^ A*, then *d_L_* (***x***,***x*****_j_**) = (*Ax* − *Ax_j_*)^T^ (*Ax* − *Ax_j_*). Since mean square error (MSE) is a function of $\hat{y}$ and $\hat{y}$ depends on ||*x* − *x**_j_***||^2^ L, MSE can be minimized by selecting an optimal value of L. 

Votes, *V**_j_***, can be replaced by *W**_j_*** as $W_{j}=exp\left(\frac{-{\left\|{AX}_{q}-{AX}_{j}\right\|}_{2}^{2}}{2\sigma^{2}}\right)$.

Finally, the GD of the error function E**_A_** with respect to the matrix *A*, which is minimized to get an optimal *A*, can be expressed as:$$\frac{\partial E}{\partial A}=2A(y_{i}-{\hat{y}}_{i})\frac{1}{\sum_{j}W_{j}}\sum_{j}(y_{j}-{\hat{y}}_{j})W_{j}(x-x_{j}{)}^{T}$$

### 3.3. From Path Kernel to kNN

**Definition** **1.**
*The tangent kernel formed for function f*
*w*
*(x) and the parameter vector v with the gradients taken at v is K*
*g*
*
_f,v_
*
*(x, x*
*’*
*) =*
*∇*
*w*
*f*
*w*
*(x) ·*
*∇*
*w*
*f*
*w*
*(x*
*′*
*).*


The path kernel associated with function *fw*(*x*) and curve *c*(*t*) in parameter space is expressed as follows:$$K_{f,c}^{p}(x,x’)=\int_{c(t)}^{}K_{f,w\left(t\right)}^{g}(x,x’)dt$$

Domingos [38] proved that $\underset{\varepsilon \underset{}{\to }0}{\lim}y=\sum_{i=1}^{m}a_{i}K\left(x,x_{i}\right)+b$ if the model y = *fw*(*x*)*,* where *f* is a differentiable function of *w*, is acquired from a training set {(*xi*, *yi**)}*mi* = 1 via GD with a differentiable loss function *L* = ∑*I L*(*yi**, *yi*); learning rate ε and *b* is the initial model. *K*(*x*, *xi*) is the path kernel for *fw*(*x*) and the path obtained by the parameters during GD, and *ai* is the average negative ∂L/∂yi traveled through the path weighted by the respective tangent kernel.

This is different from typical kernel machines since *ai* and *b* depend on *x*. At the same time, *ai* acts similarly to the example weights in the usual Support Vector Machines (SVMs) and the perceptrons, i.e., the samples for which the loss is affected by a higher degree have a greater weight. *b* is just the existing model, and the final model is computed as the sum of the existing model and the one learned via GD. The query point occurs in GD only via kernels. Since the Domingos theorem applies to every *yi* as a query in the course of GD, the training samples also occur in the model only through kernels outside of the existing model. 

In the conclusion of this section, we repeat that since both a DNN and a kNN implement versions of GD, it is natural to link them in our consecutive architecture.

## 4. Meta-Reasoning, Meta-Learning, and Introspection

The meaning of the term “meta-reasoning” is to “reason about reasoning”. Analogously, meta-learning is learning how to learn, how to perform generalization properly, how to construct training datasets, and so on. ML system should be capable of reasoning about its own operation (this is different from performing object-level reasoning). A system capable of meta-reasoning may be able to reflect and introspect, transitioning from meta-reasoning to object-level reasoning and vice versa [39].

Meta-learning helps ML developers identify algorithms that generate optimal predictions from datasets. Meta-learning refers to learning algorithms that learn from other learning algorithms. Meta-learning algorithms use learning algorithm metadata as input. They then make predictions and provide information about the performance of these learning algorithms as output. Meta-learning, in particular, discovers how to best combine the predictions from other machine-learning algorithms in the field of ensemble learning.

A split of processing into two levels has been broadly used in a Theory-of-Mind (ToM) system so that a machine can reason about itself, control itself, and actively learn. ToM is essential where some members of the human–machine team reason and learn about reasoning and learning about other team members. ToM has mainly been leveraged for the control of deduction; meta-knowledge allows navigation of a solution space leveraging heuristic rules. Meta-learning has been used to design introspective systems [40,41] which are capable of updating their own processing capabilities by tracking their status and their own internal maps of processing. DNNs cannot introspect over themselves, but their meta-reasoning components can. Meta-learning drives the design of ToM processing, where active learning and introspection occur within a unified framework [41]. In a meta-learning approach, a team member or an ML agent is represented in a meta-theory and a set of base-level (or object-level) theories, which represent the team member’s own knowledge, intentions, and beliefs, and possibly, that of the other team members. In the meta-theory, a team member or an ML agent asserts facts about the object-level theories and often performs deductions on the basis of knowledge represented in them. This requires the definition of symbols that represent, in the meta-theory, the objects of the theories.

Meta-predicates are employed to define similarity for ML in a more general form, combining the similarity of numerical expressions and the similarity of logical structures. For certain objects, if their similarity cannot be expressed in the language object (by neither a DNN nor a kNN), such a similarity can be expressed in meta-language.

The NL can be encoded by a logical language at two levels. The object level is the level of semantics. The meta-level is the level of discourse [8]. Discourse meta-expressions such as discourse or rhetorical relations take as arguments the descriptions of the description of the object-level semantic formulas. An example of such may be expressed as follows:*discourse_relation (meaning_representation(text1), discourse _relations (meaning_representation (text2)),*
where *discourse*_*relation* is a meta-predicate whose arguments range over the semantic expressions. Meta-level rules, such as the discourse rule of text organization, manipulate a representation of object-level knowledge, such as chunks of semantic knowledge, in particular, encoded by Abstract Meaning Representation [42].

If a DNN transformer acts on an object level, handling syntax and attempting to represent semantics, its discourse acts at the meta-level, interfacing with the users of its transformer model.

There is a simple example of object level and meta-level for the syntax of natural language. A word in a language, such as a *dataset,* is different from a name _dataset_; the word serves the purpose of denoting an entity, and a name denotes the word as a symbol so that we can say that “_dataset_” is composed of seven characters, is expressed in English, and its translation into Spanish is ‘conjunto de datos’. Meta-level discourse tackles the names of predicates, their arguments, and their values to formulate the rules for how text fragments can be manipulated to logically construct a plot. This construction is essentially a theory of discourse. 

### 4.1. Meta-Reasoning and Introspection

Meta-learning is valuable for improving learning performed at the semantic level, suggesting that these two levels should be interconnected. Semantic (DNN and kNN) and discourse levels interact by passing the control to each other. At the object level, the operation of *referentiation* converts a semantic expression into a name, and this name is processed by the discourse component. The inverse operation is called *introspection*; the semantic component gives control back to the meta-level discourse component. Lots of errors are made by a DNN because it lacks this introspection feature, for example, in managing dialogues. Analogously, human team members become conscious (at the meta-level of mind) of the mental states that they are currently in (at the object level). To write texts, human team members proceed from the local content (semantics) to the overall document structure and back. Analogously, ML systems with peer human users must be conscious of their success in training, active learning, selection of training methods, and self-evaluation. 

### 4.2. Linguistic Meta-Interpeter

We formally define the predicate *possible_to_predict* whose first argument is the representation (name) of an object-level semantic theory, AMR, and the second argument is a goal A. *possible_to_predict(“DNN”, “A”)* tells us that the goal *A* is expressible (provable) in theory DNN as the set of all possible DNN predictions.

Reasoning about knowledge expressed in a document can be performed at the metalevel of discourse with a call to *possible_to_predict*, and the DNN object level is simulated by providing *possible_to_predict* with a suitable description “DNN” of a semantic knowledge representation “theory” DNN.

The rules for *upward and downward reflection* are as follows:$$\begin{array}{c} T\;|{—}_{DNN}A\\ ----------------\\ Pr\;|{—}_{Discourse}\;possible\_to\_predict(“DNN”,\;“A”)\\ Pr\;|{—}_{Discourse}\;possible\_to\_predict(“DNN”,\;“A”)\\ ----------------\\ T\;|{—}_{DNN}A \end{array}$$
where |—_Discourse_ is read as provability (an ability to express) at the metalevel of learning discourse, and |—_DNN_ means provability at the object level of a DNN (what is called a neural network inference).

A metalevel clause to reason about ontological *is_a* relation is needed: *possible_to_predict(“is_a”(A, “B”)*|— *possible_to_predict(A(B)).*

If we have an ontology *mammal (“wolf”), animal(“wolf”), mammal(“dog_fido”)* and a user issues a query ‘*is dog_fido a mammal?’*, it can be represented as *is_a(“dog_fido”, “mammal”).*

We may also rely on metaprogramming at the discourse level to define properties of semantic relations as follows:

*possible_to_predict(A(B,C))*|— *symmetric(A), possible_to_predict(A(C,B)).*

*possible_to_predict(A(B,C))*|— *transitive(A), possible_to_predict(A(B,W)), possible_to_predict(A(W, C)). possible_to_predict_not(A(B,B))* |*— irreflexive(A).*

Synonyms for predicates and for their arguments can also be written as follows:

*possible_to_predict (A(B,C))*|— *symmetric(A), possible_to_predict (A(C,B)).*

*possible_to_predict (A(B))*|— *synonym(A,A1), express(A1(B)).*

*possible_to_predict (A(B))*|— *synonym(B,B1), possible_to_predict (A(B1)).*

*possible_to_predict_not(A(B))* |— *antonym(A,A1), possible_to_predict (A1(B)).*


*symmetric(“synonym”). symmetric(“antonym”).*


Hence, the knowledge base contains the following:


*synonym (“big”, “large”). synonym(“dog”, “wolf”). big(dog).*



*antonym(“angry”, “kind”). kind(wolf).*


We can infer *large(wolf*). *not angry(wolf).*The definition of similarity can be rewritten for the following pairs:
*possible_to_predict (A(B,C))|— symmetric(A), possible_to_predict (A(C,B)).*

*possible_to_predict (A(B,C))|— equivalent_pair ((B,C), (B*
*1*
*,C*
*1*
*)), possible_to_predict (A(B*
*1*
*,C*
*1*
*)).*
*equivalent _pair ((A,B), (A1,B))*|— *equivalent (A,A1).**equivalent _pair ((A,B), (A,B1))*|— *equivalent (B,B1). symmetric (“same”).*

Then, the knowledge base can be expressed as follows:


*equivalent(”cloud_run”, “cloud(process1)”). run_time(cloud_run, 70s).*


?-run_time*(cloud(process1), A)* gives *A* = *70s*.

Meta-learning helps in building embeddings relying on defining similarities at the meta-level. If the predicate *q* is defined similarly to a predicate *p*, then the system may derive the passage that includes *p* when an association of any answer with the requested predicate *q* is not available.

*possible_to_predict(A)* |— *attenuate (A,B), possible_to_predict(B).**attenuate (A(W),B(W))* |— *super_entity(A,C), super_entity(B,C).*Then, if *super_entity(“misspelling”, ”misrepresentation”)*, we obtain the following: 


*super_entity(“homonym”, ”misrepresentation”). misspelling(Word, ‘real-reel’).*


?-*homonym(Word, Q)* gives *Q* = *‘real-reel’.*

Such a similarity can also be used for transfer learning at the object level.

Meta-learning allows for a convenient formulation of common abstract queries about learning session executions without knowing exactly what this ontology is or without knowing which kind of information is available in an ontology. For an ontology that includes properties of an individual (job applicant) *man(Peter), engineer(Peter), master_degree(Peter), beginner(Peter)...* as well as desired properties from a job description *beginner,* one can formulate these into a query in the meta-language as follows:


*possible_to_predict(A(“Andrew)), job_description(beginner(man, A)).*


Analogical meta-reasoning can be implemented for reasoning about human–machine teams using the meta-predicate *has_property* as follows:

*possible_to_predict(“has_property”(“x”, B)*|— *similar (x,z), possible_to_predict(“has_property”(“z”, B).*

For a clause *has_property(Perri, “music_background”), similar(Perri, Mary)*, we can arrive at the fact *has_property(Mary, “music_background”)*.

Analogical reasoning may also occur as a transfer of properties by determination rules as follows:

*possible_to_predict(A(C,L))*|— *determine (B, A), possible_to_predict(B(C,N)), possible_to_predict(B(P,N)), possible_to_predict(B(P,L)),*

*possible_to_predict(A(B,C))*|— *equivalent_pair ((B,C), (B1,C1)), possible_to_predict(A(B1,C1)).*

### 4.3. Meta-Learning Controls Active Learning

Scholarly studies on human metacognition [43] and active learning [44] have delved into various facets of human cognition that are pertinent to meta-reasoning. These studies have examined how individuals possess an understanding of their internal states, such as the accuracy of their memories or the confidence in their judgments, and how they can make intelligent decisions on acquiring additional data samples to extend the training dataset on demand. However, supporting ML, meta-reasoning goes beyond these specific areas and encompasses the broader process of selecting or discovering the cognitive procedures that will be employed to solve a given task.

A notion of ‘metalevel rationality’ [45] can be leveraged by the meta-level to actively learn. Meta-level rationality is assessed by how well the algorithm follows shaped-charge learning in order to select the best dataset expansion and other learning actions trading off expected utility with the costs of taking more time and expending more computation before acting. From this perspective, rationality is not just about making good decisions and drawing good inferences but also about employing efficient cognitive strategies.

Figure 5 illustrates a meta-level Markov decision process. The initial meta-level rewards capture the cost of computation, and the final meta-level reward captures the benefits of computation by the expected object-level reward for choosing a *cognitive* action based on the final belief state.

We introduce the meta-reasoning value of computation and acquiring a new data sample (VOCADS) based on current beliefs as follows:$$VOCADS=E_{p(b^{’}|b,c)}[{max}_{a^{’}}E\left[U\left(a^{’}\right)|b^{’}\right]-{max}_{a}E\left[U\left(a\right)|b\right]]-cost(c),$$
where *c* is a computation (running a DNN session with a currently available dataset that matches the current belief), *b* is the overall system’s current belief, *b’* is the updated belief resulting from executing a DNN learning session (computation) *c*, and $\left[U\left(a\right)|b\right]$ is the expected gain of taking action an over the distribution of outcomes corresponding to belief b.

Meta-learning over (DNN → kNN) is expected to perform the learning session with the highest VOCADS, or, if no learning session has positive VOCADS, give up any expectation of prediction accuracy improvement at all. The limitation here is that computing the VOCADS itself is computationally expensive as it requires executing each computation to select the optimal cognitive action.

Ackerman and Thompson’s (2017) [46] framework of monitoring and controlling reasoning adapted to shaped charge architecture is shown in Figure 6 for the case of the question-answering system.

The left side represents the object-level processes involved in reasoning, considering that various reasoning theories make different assumptions about the timing and nature of those processes. The middle area specifies the reasoning and learning monitoring processes, and the right side provides the associated control functions, including VOCADS-based cognitive action selection. All monitoring processes reflect the shaped-charge system’s assessment of the probability of success or failure in each task before, during, or after engaging in the task. These assessments trigger a variety of control decisions, including taking cognitive action, allocating time and effort to a task, and choosing a strategy to complete the task.

A reasoner is first expected to make an initial Judgment of Solvability [47], which contains the reasoner’s assessment that the problem is solvable. This Initial Judgment of Solvability decides whether to attempt a solution, give up, seek external help, and so on. In terms of question answering, the system needs to decide if an answer for a given question exists and needs to be retrieved or has to be formed via a generative model. 

According to [47], the ‘Metacognitive Reasoning Theory’ elucidates the connection between the monitoring and control of reasoning. This theory specifically addresses situations in which the problem’s context triggers an immediate, initial response. This initial response is believed to encompass two aspects: the answer itself and a subjective Feeling of Rightness associated with it. When the Feeling of Rightness is intense, it serves as a signal that additional reconsideration is unnecessary. As a result, reasoners dedicate minimal time to re-evaluating their answers and are unlikely to alter their initial stance [48].

Opposingly, a weak Feeling of Rightness is accompanied by longer periods of reconsideration and a higher probability of changing answers. Importantly, because Feelings of Rightness are derived from cues that may be poorly correlated with accuracy (see next section), reasoners may be led to wrongly accept their initial intuitions with little reconsideration.

Under meta-control, we rely on an uncertainty sampling approach to active learning, where we acquire data samples for which the current model is least confident in its predicted label. The outputs of the model *M* are defined as a probability distribution over possible labels *p*(*y* | *x*; *M*), and the cognitive action for data sample selection is defined as follows:$$a\left(x;M\right)=-{max}_{i}p(y_{i}|x;M)$$

We illustrate the capability of meta-controlled active learning with the following example (Figure 7). Here, the available initial training data does not provide the model with a clear set of key features: is it the shape or the color of the object? Pretraining enables models to identify and weigh various rich features, eliciting labels from informative examples such as green squares that clarify the user’s intention.

### 4.4. Obtaining Meta-Learning Structure of DNN Engines

Traditionally, discourse-level analysis controls the overall structure and flow of information being communicated in a text. If a text is subject to analysis via an ML system, then a control level similar to discourse organizes and controls processing units for this text. For example, answering a question might include the following tasks: Entity extraction from a question and answer;Sentiment analysis of occurrences of these entities in answer;Entity occurrence coordination between a question and an answer.

The overall structure of these tasks follows the directed acyclic graph or tree. It turns out that the structure of processing tasks can be viewed analogously to the overall thought structure being communicated by a text. While a discourse tree of text is a well-established formalism, a discourse tree of processing/extraction units is an abstraction for the meta-learning of text we propose in this study. Forming a high-level representation of text, we try to abstract away from whether humans understand a sequence of elementary discourse units or whether a machine applies processing units to this text. Text organization is expected to be invariant with respect to whether humans or machines process it; the overall structure of cognition is what matters. 

Let us try to form a meta-learning structure for text generation in a biography domain. We ask ChatGPT to generate a fragment (answer a question) about Alexander Pushkin’s father (the father of a famous Russian poet of the 19th century) as an entity. An instance of ChatGPT results in what may look like the following:

Q: Where and when was Pushkin’s father born? 

A: Alexander Pushkin’s father, Sergei Lvovich Pushkin, was born on 26 May 1767, in Moscow, Russia. He was a member of the Russian nobility and served as a colonel in the Imperial Russian Army. Sergei Pushkin married Natalia Alexandrovna Goncharova, Alexander Pushkin’s mother, in 1795. Alexander Pushkin was the oldest of their six children. Sergei Pushkin died in 1848, at the age of 81.

The fact is Natalia Goncharova is Pushkin’s wife, not his mother. How can we systematically avoid this error? We need a kNN-based verification to perform a fact check of each sentence by finding the nearest neighbor on the web. For a sentence where Pushkin’s mother and wife are confused, our fact checking can identify and correct the failure by looking up the sentence’s neighbor on the web (Figure 8).

It is explicitly expressed by the nearest identified neighbor that Natalia Alexandrovna Goncharova is the wife, not the mother of Alexander Pushkin. So, the sentence “Sergei Pushkin married Natalia Alexandrovna Goncharova, Alexander Pushkin’s mother, in 1795” either needs to be removed or updated. In both cases, a discourse structure can be damaged, so we need to identify entities and relationships between them to decide on which entity or relation to substitute to maintain the truth. 

We will automatically form the text analysis structure from the discourse representation of biography sketches. In this particular case of biographical texts, rhetorical relations are trivial (*Elaboration* and *Joint*) as there is neither contradiction, causal links, nor other nontrivial relationships between discourse units. However, there are interesting relationships between entities that need to be identified (Figure 9).

Now, we establish a structure for DNN detectors which are required to perform fact-checking. We need entity extractors and relation extractors for that, and the above entity structure yields a meta-learning structure for these DNN detectors.

Figure 10 establishes a meta-learning structure for this individual text. For multiple texts of the same genre, we generalize across multiple meta-learning structures for individual texts and obtain a set of configurations of DNN engines. As a result, we obtain a meta-learning structure suitable to process text of a specific genre. A meta-learning structure can be implemented as a directed acyclic graph of processes, such as with [49]. In our previous studies, we described graph generalization in detail for syntactic, semantic, and discourse representations. For the meta-learning structure, we leverage discourse representation in the form of a discourse tree [50] or entity occurrence graph [51,52].

Apache Airflow is an open-source platform used to programmatically author, schedule, and monitor workflows [53]. Developed by the Apache Software Foundation, Airflow has become an increasingly popular tool for managing and automating data engineering pipelines. Airflow is composed of two components: (1) a web interface used to configure and monitor workflows and (2) an underlying execution engine that runs the workflows. Airflow is highly extensible and allows users to customize workflows with Python code. This allows complex logic to be incorporated into workflows and helps to automate complex tasks such as data transformation and machine learning model training. Airflow also provides powerful features such as task scheduling, logging, and exception handling. Airflow is used by many organizations to manage and automate data engineering workflows. Its powerful features make it a popular choice for managing complex data pipelines. Airflow is also easy to use and can be quickly set up and deployed. Additionally, Airflow can be integrated with other popular software, such as Hadoop, Spark, and Kubernetes, making it a very versatile tool. Overall, Apache Airflow is a powerful and versatile platform for data engineering and automation.

We now outline a meta-learning DNN engine structure construction algorithm (Algorithm 1):
** Algorithm 1** Meta-learning structure construction**         Input:** (1) A corpus of texts          (2) Initial chain of processing components         **Output:** A set of optimal meta-learning structures (a task pipeline graph)(1)Build a syntactic parse tree, entity graph with coreference, discourse tree for each text in corpus;(2)Select a path in each such tree which better matches the Initial chain of processing components:(a)Find a mapping between each rhetorical relation and entity-entity relation→ task result propagation; and,(b)Form a reduction of an elementary discourse unit into an input text fragment for each processing component.(3)Expand the chain of processing components to other paths in each tree; and,(4)Align all trees and obtain the set of the most representative trees (cluster centers of trees).                As to the algorithm to apply the meta-learning structure to a given text:         **Input:** (1) **A text**
         (2) A set of meta-learning structures          **Output:** A prediction based on the chosen meta-learning structure

(1)Iterate through each structure and obtain the prediction;(2)Identify processing structure with the highest confidence for prediction;(3)Apply reinforcement learning to this selected fixed processing structure to optimize the weight for each participating predictor (such as in an ensemble classifier), if applicable.

Figure 11 gives an example of processing components that are the subjects of the above algorithm.

## 5. Shaped-Charge Architecture

### 5.1. From Fine Tuning to a kNN Extension to Simulate Low Structural Entropy

Even though pre-trained language models are more robust in terms of out-of-distribution generalizations than previous models [54], they are still not performing well in domains significantly different from the ones they have been pre-trained on. Adaptive fine-tuning helps to tackle such a shift in distribution by employing meta-learning to force fine tuning the model on data that is closer to the distribution of data in the target domain. In particular, meta-learning makes sure the model is fine-tuned on additional data prior to fine-tuning the specific problem to be solved, which can be seen below. Adaptive fine tuning only requires unlabeled data. A masked-language model (MLM) is used for pre-training. Our DNN component can be fine tuned, for example, as depicted in Figure 12. 

### 5.2. Meta-Learning over (DNN → kNN)

We finally approach the overall architecture for wrapping a DNN. At training time, training datasets are selected and maintained via a meta-learning component. Based on the current prediction results, meta-learning decides which additional data samples to involve (active learning). From the discourse structure of sample texts, meta-learning automatically discovers how to combine individual DNN processing components. Meta-learning also transfers the trained model to a similar problem domain if its evaluation is successful in it (see Figure 13).

In the inference time, a prediction made by a DNN is confirmed or rejected by a kNN component. Meta-learning verifies the prediction made by the DNN by maintaining a current set for a kNN and maintaining a threshold. Even if the current prediction is not modified by the kNN, the identified neighbor is used for an explanation of why the input belongs to a certain class. The final prediction is computed as meta-learning assesses the confidence level, given the results of the DNN and kNN. 

In the following section, we apply this architecture to the domains of question answering, summarization, and content generation, instantiating each component of the proposed architecture.

## 6. Discussion

We linked a DNN with a kNN via GD, inspired by the path kernel formalism. For linear DNNs, the path kernel reduces to the dot product of data samples. It has been understood five decades ago that a single-layer perceptron is a kernel machine, with the dot product as the kernel [55]. This result can be interpreted as a generalization of this to multilayer perceptrons and other models. It is also related to [56] proof that Hopfield networks, a predecessor of many DNN architectures, are equivalent to a kNN, a predecessor of kernel machines, with Hamming distance as the distance function. 

The employed path kernel representation assumes that the learning rate is sufficiently low for the trajectory of the weights during GD to be well interpolated by a smooth curve. This is a typical assumption in the GD analysis and usually works well in practical applications since the learning rate should be low to avoid a divergence situation [57]. It is not very clear to what extent GD models can still be approximated by kernel machines under a high learning rate. 

Bakhtin et al. (2022) [58] describe Cicero as an AI agent that is close to human-level performance in Diplomacy, a strategy game that is based on cooperation and competition with a focus on NL negotiation and tactical coordination. Cicero integrates an NL with planning and reinforcement learning algorithms by reasoning about players’ beliefs and intentions from their conversations and then generating dialogue supporting the goals of players.

Cicero predicts the most probable human actions for each player given the board’s state and the conversation between players, using both as the starting point for a planning algorithm using reinforcement learning. The output of planning is an action for the agent as well as intentions and beliefs concerning other players’ actions. This output is a basis for intent selection for the conversation support component to be conditioned on. Generated messages are subject to multiple filtering steps before a final message is produced (Figure 14).

Deep learning pioneer Schmidhuber (2022) [59] votes for an increased focus on meta-learning, an automated combination of multiple learning mechanisms with different aptitudes across different tasks. Clune (2020) [60] also advocates for meta-learning, with a more evolutionary twist expressed by concerns about how bad actors might use artificial general intelligence and by arguing that addressing such potential misuse was among the most important questions facing humanity.

Modern AI models are intended to be run on computers but are used in human-driven team-based applications. This creates an explicit mismatch between AI forms of processing and human ways of discovering and managing knowledge. Dudyrev et al. (2022) [61] introduce a new concept of “Human Knowledge Models” designed to focus on the computational abilities of human team members. Relying on an extensive corpus of cognitive research, the authors formalized the definition of Human Knowledge Models into a special form of ML. Then, by training the models with human processing capabilities, it became viable to acquire human-like knowledge that human team members can not only understand but also process.

Many AI scientists believe that when artificial general intelligence becomes plausible, large language models such as GPT-4 may be considered only as a part of the solution. Scaling up a DNN alone until they absorb the entire internet is useful only to a degree [62]. Trustworthy, general artificial intelligence for team collaboration is expected to come from architectures that rely on more structured components with more internal knowledge, including tools for reasoning and planning. *Meta-learning over (DNN → kNN)* is expected to be a step in this direction, integrating large LMs with a broad spectrum of other techniques.

We believe that our approach is not only a theoretical investigation but also a practical ML tool. Its efficiency and computational cost are critical points of modern DNN architectures since they require a lot of resources to train and even to run on a single machine. However, a lot of effort is put into making DNN components more accessible and easy to integrate. Meta-learning over (DNN → kNN) is targeting the same goal because it does not increase the overall DNN learning cost but instead gives human users reasonable insights about the overall system quality.

Modern DNN methods for NLP tasks have become increasingly popular due to their cost and scalability. With powerful GPUs becoming more readily available, it has become easier to train complex deep-learning models with large datasets. Moreover, deep-learning libraries such as TensorFlow, Pytorch, and Keras have made it simpler for users to apply deep learning to their projects. This, combined with the lack of computational overhead that comes with deep-learning methods compared to traditional methods, has helped to make deep learning more cost-efficient.

These cost savings also extend to scalability. The same deep-learning models can be used to process large amounts of data with relative ease. This makes it a powerful solution for dealing with large datasets while also providing greater accuracy and faster training time than traditional methods. Additionally, modern deep-learning models are easily able to leverage parallel computing to speed up training and inference time.

Shaped-charge learning captures the idea that generalization should occur at multiple levels of abstraction. This idea has been widely used in Bayesian models of cognition. For example, hierarchical Bayesian inference can be used to learn about the properties of objects that words tend to label (such as shape) at the same time as learning the meaning of individual words [63] and to learn about the kinds of causal relationships that exist at the same time as learning those relationships. While Bayesian inference generically indicates how a learner should combine data with a prior distribution over hypotheses, a hierarchical Bayesian model learns the prior distribution through experience.

One of the novelties of shaped-charge learning is that its meta-learning controls its active learning. Ref. [64] discovered that better active learning is an emergent property of the pretraining process; pretrained models require up to five times fewer labels when using uncertainty-based active learning, while non-pretrained models see no or even negative benefits. These boosts in performance come from an ability to select examples with attributes that disambiguate the intended behavior, such as rare product categories or atypical backgrounds. These attributes are far more linearly separable in the pretrained model’s representation spaces vs. non-pretrained models, suggesting a possible mechanism for this behavior. Meta-learning facilitates finding these atypical data samples.

kNN allows us to stay as close to the foundation of ML as possible. A British philosopher John Stuart Mill presented and analyzed five techniques of experimental investigation in his notable work, the System of Logic [65]. These approaches, known as the method of agreement, the method of difference, the joint method of agreement and difference, the method of residues, and the method of concomitant variation, were extensively deliberated upon by Mill. He asserted that these methodologies serve as the means to uncover and establish causal connections, playing a crucial role in scientific exploration. Mill referred to these approaches as the “eliminative methods of induction”. Methods of agreement can be expressed via kNN operations.

To illustrate the fundamental nature of Mill’s experimental inquiry techniques, we can examine the two simplest methods: the method of agreement and the method of difference. Mill’s principle for the method of agreement is as follows: “When multiple instances of the phenomenon being studied share only one common factor, that particular factor is the cause (or effect) of the observed phenomenon”. ‘Sharing one common factor’ can be expressed as being neighbors, where similarity includes a representation of this common factor.

For example, if a number of people who are suffering from a certain disease have all spent significant time without sunlight but have, in other respects, had quite different diets, lived in different conditions, belong to different races, and so on, so that the lack of sunlight is the only feature common to all of them, then we can conclude that the lack of sunlight is the cause of this particular disease. The similarity between instances of health data is expressed by possessing a ‘lack of sunlight’ feature. Hence, kNN can express this method of agreement proposed by Mill (as well as the method of disagreement and other ML foundational methods).

Multiple projects have demonstrated that transformers are fairly robust to pruning. However, it has been discovered [66] that pre-trained transformer encoders are very sensitive to the removal of a very small number of features in the layer outputs (below 1/10,000% of model weights). In the case of BERT and other pre-trained encoder transformers, the affected component is the scaling factors and biases in the LayerNorm. This observation is due to an interaction of high-magnitude scaling factors and biases in the same dimension throughout the model rather than magnitude alone. It emerges early in the training and systematically distorts the embedding space.

To adjust BERT for out-of-distribution data where a set of specific features unseen in the test data can ruin the performance, shaped-charge comes to the rescue by identifying such cases, actively learning them, and/or overwriting predictions made by stand-alone DNN. The meta-level of a shaped charge can identify cases and features of wrong prediction and perform disabling on outlier weights, which is essential to address the pruning issue.

Disabling LayerNorm weights, facilitated by the meta-level, sets the outlier weights *a* and *b* of the output LayerNorm to zeroes (Figure 15). This results in the masking of the corresponding features in the output vectors. This operation is repeated for all transformer layers of the encoder.

The shaped charge can also control model compression. As BERT’s largest weights form subnetworks of a network of all weights, it can be retrained alone to reach a performance close to that of the full model. 

The use of machine-learning models in human–machine teams has the potential to increase the efficiency and accuracy of certain tasks. However, this technology also brings a number of limitations and potential ethical implications that should be considered before its implementation. One of the primary limitations of machine-learning models is their lack of interpretability. This means that, while they can produce accurate results, it is often difficult to understand why they came to a particular conclusion. This lack of interpretability can lead to errors and potentially lead to a lack of trust in the model or its outcomes. In addition, machine-learning models may also be limited in their ability to adapt to changing circumstances, making them unsuitable for certain tasks or environments. For example, a model trained to recognize objects in a certain environment may not be able to perform properly when faced with a different environment. Finally, the use of machine-learning models in human–machine teams raises potential ethical implications. For example, if a machine-learning model produces results that could harm people, it could be seen as unethical. Another potential ethical implication of using machine-learning models in human–machine teams is the potential for privacy violations. Machine-learning models are often trained on large datasets that contain personal data. One of the most pressing ethical issues is the potential for machine-learning models to result in biased decisions, especially when used in decision-making roles. If a machine-learning model is trained on data that is biased in terms of race, gender, or any other protected characteristic, it is likely to make biased decisions. This could lead to serious ethical issues, such as unfair treatment of individuals or groups. As such, it is essential to ensure that machine-learning models are trained on unbiased data and that any potential biases are identified and addressed.

## 7. Conclusions and Future Work

We conclude that extending a DNN with a kNN and performing meta-learning over this extension significantly is expected to significantly improve the overall accuracy and explainability of ML as well as human–machine team support. An improvement in performance is demonstrated in question answering, summarization, and content generation work in the second paper of this series. In all of these domains, the ML component will need to become a supporting part of human–machine teams, be integrated into the teamwork seamlessly, with low structural entropy, and efficiently (low wasted energy), and help the team to achieve producing maximum entropy production.

There are the following advantages of shaped-charge learning in comparison with other hybrid *DNN + statistical/logical ML* architectures:(1)kNN’s role is superior for explainability as it is the closest to the logical foundation of induction as a basis of learning. Such advantages of stand-alone kNN include computational efficiency, simplicity to interpret, usability for both regression and classification, and high accuracy—these are all leveraged by shaped-charge learning.(2)A consecutive *DNN → kNN* approach leads a learning session via step-by-step zooming in on a correct prediction in a search space which is initially huge. It allows for overcoming the limitations of stand-alone DNN and stand-alone kNN.(3)The meta-support of the shaped charge implements full control over which method is applied to which kind of data at each decision step. The meta-level support of gradient descent in *DNN → kNN* assures an effective combination of manually selected features for kNN with auto-feature engineering via DNN. A fine-tuning of DNN is implemented by meta-learning controlling active learning.(4)Shaped charge learning can be naturally integrated with rule-based systems and classical object-level reasoning components such as reasoning about space and time, mental states, physical processes, and other domains which can be encoded as axions. These reasoning systems can complement kNN or fully substitute it.

As mentioned earlier, in our second paper, we will describe particular applications of our architecture to the domains of question answering, summarization, and content generation. For future work, we plan to explore several different directions of the proposed approach and extensions of shaped-charge architecture. We suppose that it can be implemented for some other NLP tasks such as dialogue generation and planning, machine translation, information extraction, and fact checking. Another promising direction is related to moving beyond English to low-resource languages such as Slavic [67]. The bottleneck here is DNN since our kNN component is almost language independent. However, over the last few years, DNN demonstrate spectacular results in the adaptation to the different languages working in the zero-shot or a few-shot manner [68,69]. Therefore, we believe that shaped-charge learning would be useful for a high variety of tasks and languages.

However, some limitations of the DNN still remain. Firstly, deep-learning models tend to be highly dependent on or biased toward the specific dataset used. As a result, models trained on one dataset may have little utility in transitioning beyond that dataset into more general-use applications. Additionally, deep-learning models require a very large amount of labeled data for training, which makes them difficult to use in certain settings where labeled data is limited. For example, it has been reported in certain cases that their performance decreased compared to shallow learning models in settings where labeled data is scarce. Furthermore, a related limitation is the computational cost associated with deep-learning models. This can be prohibitively difficult when the models are extremely complex, particularly in settings with limited resources. 

## Figures and Tables

**Figure 1 entropy-25-00924-f001:**
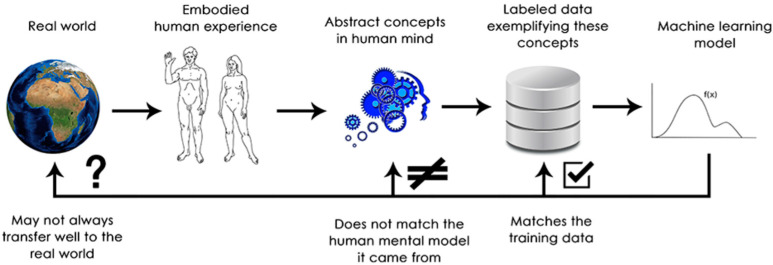
Approximation of the real world with DNN and human cognition.

**Figure 2 entropy-25-00924-f002:**
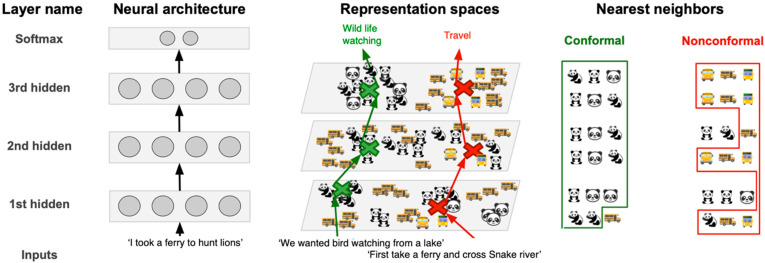
Parallel DNN and kNN architecture.

**Figure 3 entropy-25-00924-f003:**
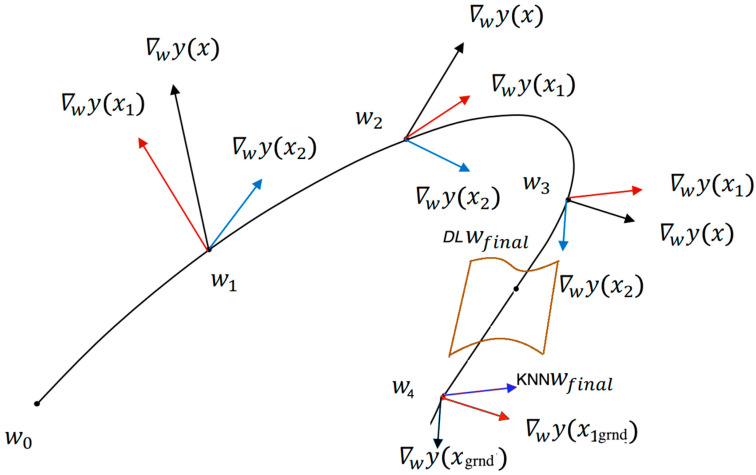
A path kernel first for a DNN and then for a kNN.

**Figure 4 entropy-25-00924-f004:**
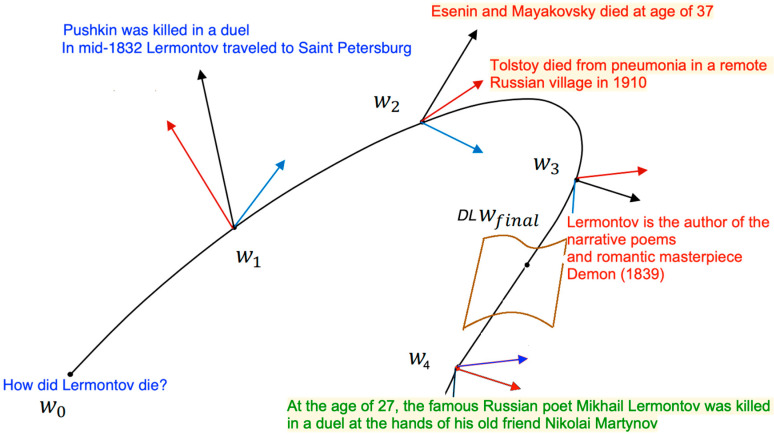
Linguistic interpretation of a path kernel in finding an answer to a question.

**Figure 5 entropy-25-00924-f005:**
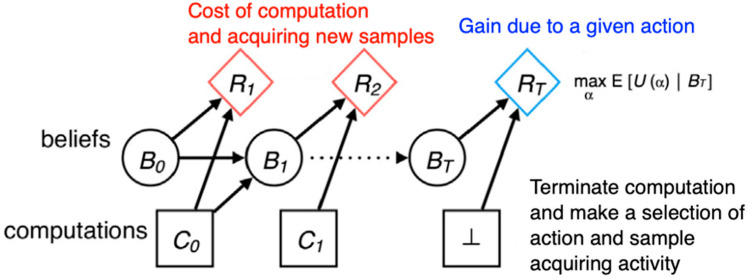
Beliefs, computations, and optimization of the meta-learning architecture.

**Figure 6 entropy-25-00924-f006:**
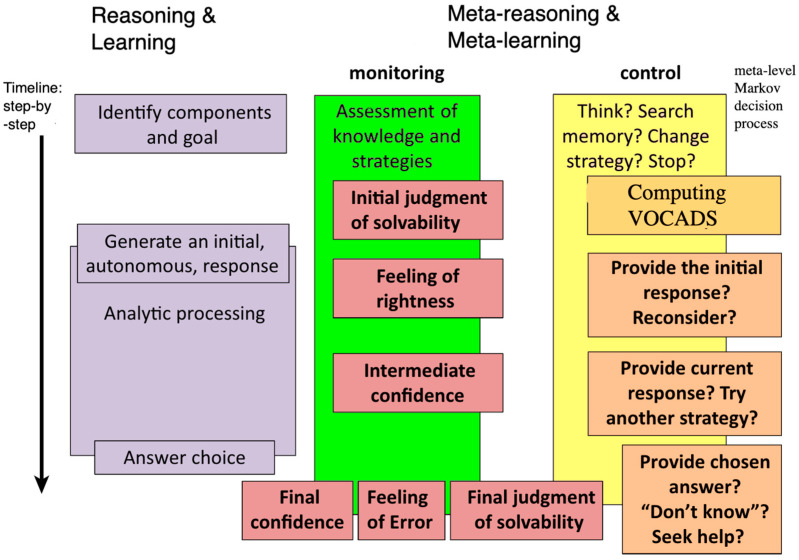
A framework for monitoring object-level learning and reasoning.

**Figure 7 entropy-25-00924-f007:**
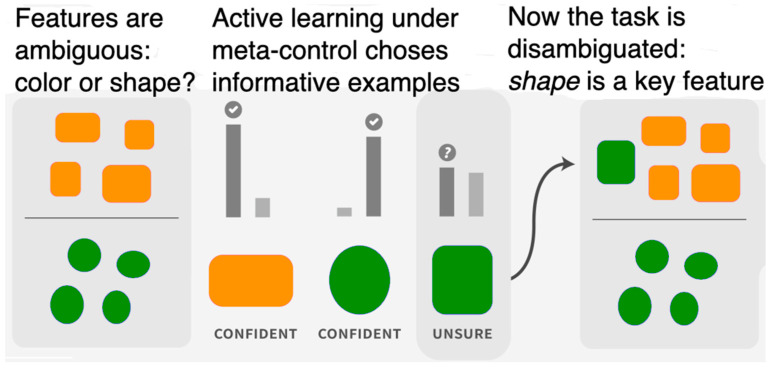
Active learning under meta-control can resolve feature selection ambiguity in a dataset.

**Figure 8 entropy-25-00924-f008:**
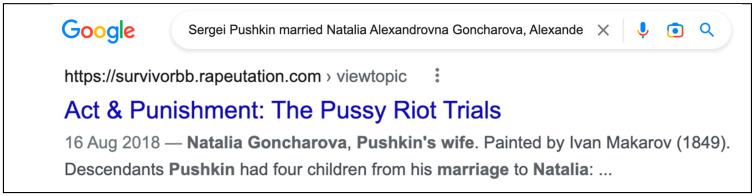
kNN for fact-checking of generated text.

**Figure 9 entropy-25-00924-f009:**
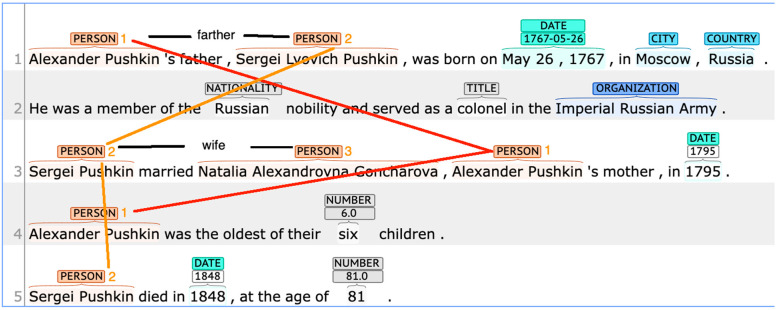
Entity graph on a parse tree for a text being fact checked.

**Figure 10 entropy-25-00924-f010:**
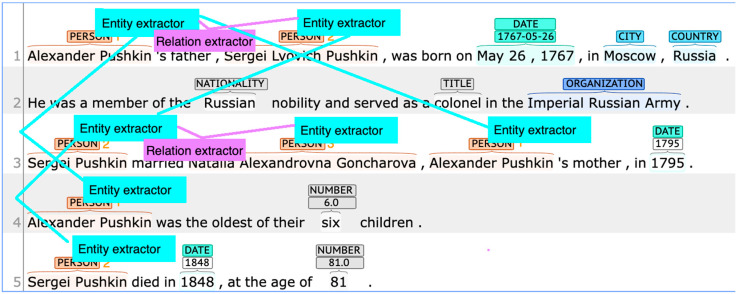
Entity graph yields the processing graph for DNN processors of entity extraction and relation extraction.

**Figure 11 entropy-25-00924-f011:**
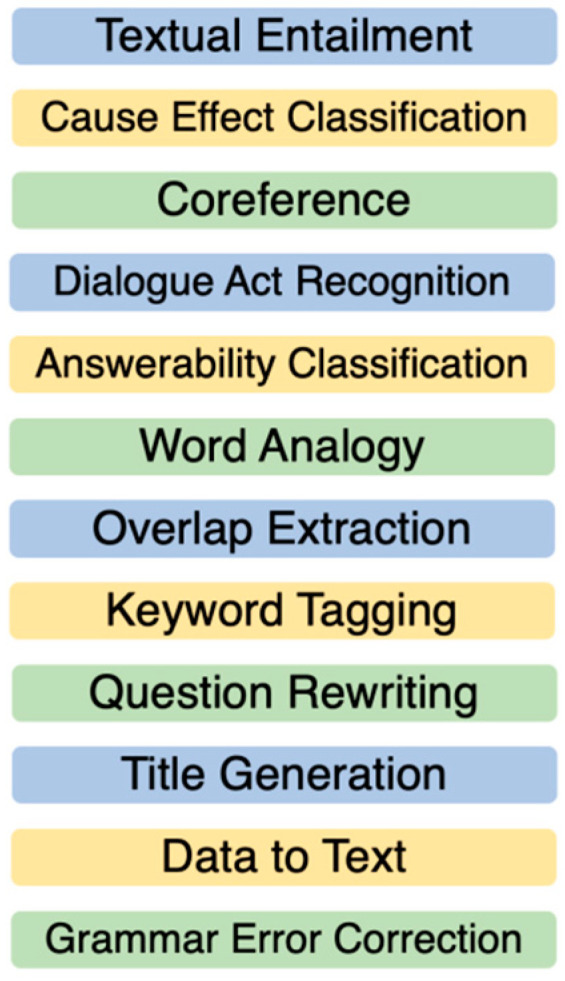
Some NLP processing components that can be used for the meta-learning DNN engine structure construction algorithm.

**Figure 12 entropy-25-00924-f012:**
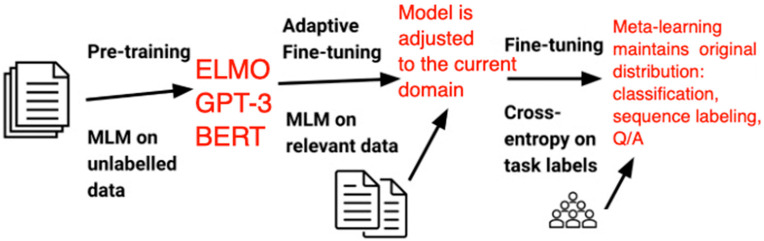
Adaptive fine-tuning architecture to simulate low structural entropy costs for the DNN component of the meta-learning framework.

**Figure 13 entropy-25-00924-f013:**
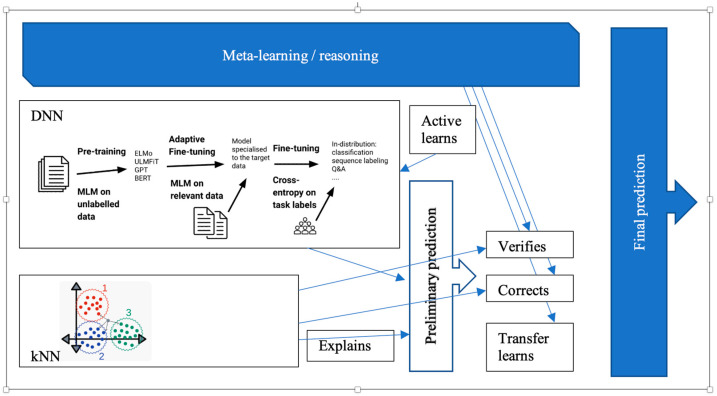
Proposed shaped charge learning architecture simulates maximum entropy production.

**Figure 14 entropy-25-00924-f014:**
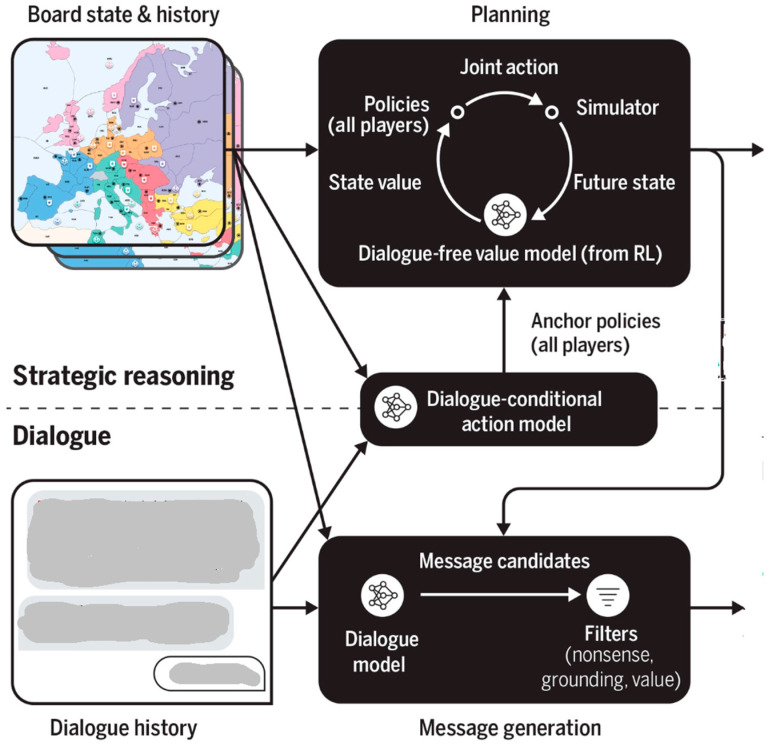
An example of hybrid reinforcement learning (RL) and reasoning (for planning) architecture.

**Figure 15 entropy-25-00924-f015:**
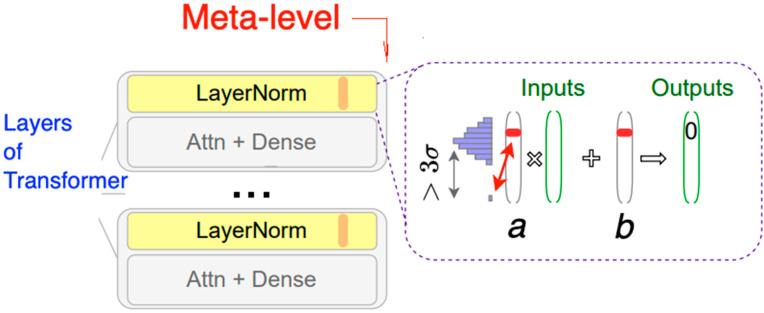
Illustration for the disabling of LayerNorm weights.

## Data Availability

Not applicable.

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
