# Peer review of "Shaped-Charge Learning Architecture for the Human–Machine Teams"

_entropy, 2023, doi:10.3390/e25060924_

Round 1

Reviewer 1 Report

The submitted paper is interesting and deals with a very current topic.

However, to enhance its value for readers, the following points require further elaboration:

- The methodology used to evaluate the proposed solution needs to be explained in greater detail. While the model holds potential for application in various areas, the authors should compare its results with those obtained using alternative methods, at least in one area.

- The choice of the kNN algorithm requires further explanation, including whether there were other algorithms considered and what advantages kNN has over them.

- The authors mention NLP tasks such as question-answering, summarization, and content generation as potential applications of the model. It is important to investigate how the model performs in languages with limited resources, such as Slavic languages (for example papers https://doi.org/10.1016/j.dss.2013.02.002 and https://doi.org/10.1371/journal.pone.0242050).

- It would be helpful to explicitly state the contributions of the proposed model in the Introduction section's concluding paragraph.

- it is necessary to consider the limits and limitations of the proposed model

Overall, with these necessary improvements, the research has the potential to make a valuable contribution to the field.

No comment

Reviewer 2 Report

This article describes a Meta-learning/DNN → kNN architecture that integrates deep learning with explainable nearest neighbor learning (kNN) to form the object-level, having deductive reasoning-based metalevel control learning process, performing validation and correction of predictions in a way that is more interpretable by peer team members. 

The proposed solution is attractive. However, reading the manuscript raised some questions. First, in the introduction, the authors should have explained the novelty of their proposed solution compared to state-of-the-art research in this field. Section 5 is very short and ends with a statement that promises the application of the proposed solution to the domains of question-answering, summarization and content generation. But there is no section that would explain the application and present some results that could be used to interpret the benefits of the proposed solution. It looks like something is missing. Also, after Figure 12, there is Figure 18. And the numbering of figures before in the text seems wrong. Some figures are not properly explained. Reading the article gives the feeling that the writing was rushed and that the article was not properly concluded.

Reviewer 3 Report

The paper proposes a new architecture called Shaped-charge learning that combines deep learning with explainable machine learning components such as nearest neighbor learning and meta-learning to improve the interpretability, robustness, and adaptability of machine learning models in supporting man-machine teams. The following are suggestions from the reviewer:

1.          It is recommended that the paper provide more detailed and quantitative results to demonstrate the effectiveness of the Shaped-charge learning architecture in the case studies mentioned.

2.          The paper could provide more background information or references to other sources for readers who are not familiar with concepts such as Apache Airflow, kNN, and DNN.

3.          A more thorough evaluation of different approaches for handling out-of-distribution data would help readers understand the advantages and limitations of the Shaped-charge learning architecture.

4.          Providing more information on the active learning algorithm used or the criteria for selecting additional samples would be helpful.

5.          It is suggested that the paper provide a more detailed comparison with other existing approaches, and more empirical evidence to support claims about the effectiveness of the Shaped-charge learning architecture in various applications.

6.          The conclusion section could be improved by summarizing the main contributions of the research in a more concise manner.

7.          The paper should provide more information on the computational cost and scalability of the Shaped-charge learning architecture.

8.          It is recommended that the paper include a more thorough discussion of the limitations and potential ethical implications of using machine learning models in man-machine teams.

9.          The references section could be expanded to include more recent and relevant studies on explainable machine learning and human-machine collaboration.

10.      Future work should be discussed in greater detail, including potential directions for further research on the Shaped-charge learning architecture, its limitations, and possible extensions to other domains beyond question-answering and summarization.

The paper requires significant revisions and improvements in terms of providing more detailed and quantitative results, background information and references, comparisons with other approaches, discussion of limitations and ethical implications, and more thorough evaluations. The paper also needs to improve the conclusion and provide more information on computational costs and scalability. The references section should also be expanded, and future work should be discussed in greater detail. Therefore, the paper is deemed to require major revision before it can be considered for publication.

Moderate editing of English language

Round 2

Reviewer 2 Report

The authors have addressed issues raised by all reviewers adequately. I agree with the third reviewer that this paper should provide more detailed and quantitative results to demonstrate the effectiveness of the Shaped-charge learning architecture in the case studies mentioned. This part was left out in the revision of the manuscript. The authors have explained that this will be presented in the following paper. The manuscript can be accepted for publishing.

Reviewer 3 Report

The authors have addressed all concerns raised. Detailed, quantitative results are provided, and substantial background information is included for clarity. Comparisons with other approaches are thorough and comprehensive. Discussion of limitations and ethical implications is insightful, and the paper now illustrates computational cost and scalability. The conclusion is concise and impactful, and future work discussion has been expanded. With these revisions, the paper presents a robust and significant contribution to the field. I recommend acceptance.

Minor editing of English language required